# TENET: LEVERAGING TESTS BEYOND VALIDATION FOR CODE GENERATION

## ABSTRACT

Test-Driven Development (TDD) is a widely adopted software engineering practice that requires developers to create and execute tests alongside code implementation, ensuring that software behavior is continuously validated and refined. In the era of vibe coding, where developers increasingly delegate code writing to large language models (LLMs) by specifying high-level intentions, TDD becomes even more crucial, as test cases serve as executable specifications that explicitly define and verify intended functionality beyond what natural-language descriptions and code context can convey. While vibe coding under TDD is promising, there are three main challenges: (1) selecting a small yet effective test suite to improve the generation accuracy and control the execution workload, (2) retrieving context such as relevant code effectively, and (3) systematically using test feedback for effective code refinement. To address these challenges, we introduce TENET, an LLM agent for generating functions in complex real-world repositories under the TDD setting. TENET features three components: (1) a novel *test harness mechanism* that selects a concise test suite to maximize diversity of target usage scenarios; (2) a *tailored agent toolset* that performs efficient retrieval of relevant code with interactive debugging; and (3) a *reflection-based refinement workflow* that iteratively analyzes failures, replenishes context, and applies code refinement. TENET achieves 69.08% and 81.77% Pass@1 on REPOCOD and RepoEval benchmarks, outperforming the best agentic baselines by 9.49 and 2.17 percentage points, respectively. In addition, this is the first study of test-driven code generation with repository-level context, examining how different aspects of test suites affect the performance of LLM agents under the TDD setting.

## 1 INTRODUCTION

Test-Driven Development (TDD) is a widely adopted practice in software engineering that tightly couples the testing and implementation processes (Beck, 2022). Rather than treating tests as afterthoughts, TDD requires developers to create and execute tests continuously throughout the development life-cycle. In practical TDD workflows, developers typically start by writing test cases that specify the desired behavior or capture potential failure scenarios, then incrementally implement and refine code to satisfy these tests. Extensive empirical studies have demonstrated that TDD improves code quality, enhances design clarity, boosts developer productivity, and supports long-term maintainability (Mathews & Nagappan, 2024; Piya & Sullivan, 2024; Tian et al., 2025; George & Williams, 2004; Williams et al., 2003; Janzen & Saiedian, 2008; Sheta, 2023; Mafi & Mirian-Hosseinabadi, 2023; Cassieri et al., 2024; Baldassarre et al., 2021).

In the era of vibe coding (Karpathy, 2025), where developers increasingly delegate code writing to large language models (LLMs) by specifying high-level intentions, TDD becomes even more crucial. As LLM-generated code may amplify ambiguities or inconsistencies in developer intent, systematically specifying and validating requirements through test cases is essential to ensure correctness, intended functionality, and maintainability. Recent industry practices further echo this view, emphasizing that TDD becomes especially powerful in the context of agentic coding (Anthropic, 2025; Cherny, 2025).

Accordingly, TDD provides a more reliable setting for developing and evaluating code generation techniques with repository-level dependencies. Existing work typically frames the task by requiring the LLMs to generate target functions based on natural language descriptions of intent and contextual

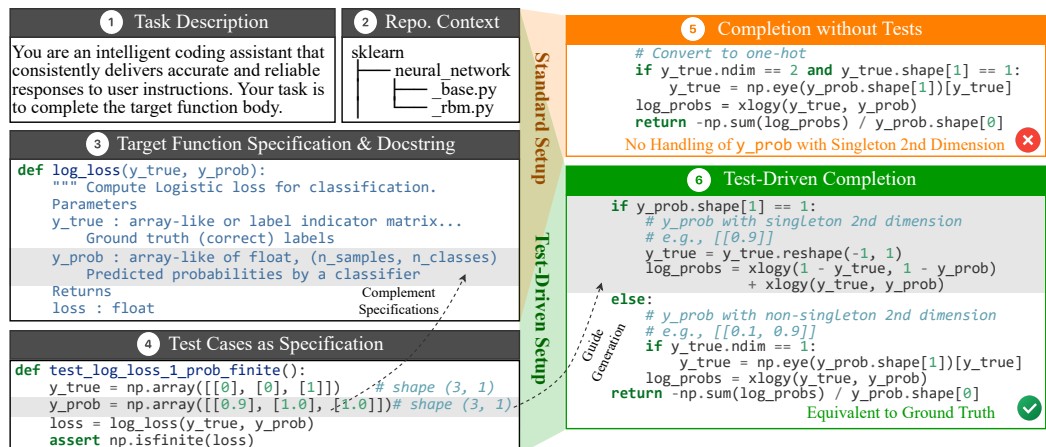

Figure 1: Examples of repository-level code generation under standard and test-driven setups.

code retrieved from the repository (Zhang et al., 2023; 2024b; Li et al., 2024). However, such descriptions and context are often insufficient to fully convey the intended functionality of the target function, making it infeasible for the LLMs to produce the expected implementation. In contrast, test cases offer executable specifications that explicitly define correct behaviors and failure scenarios.

Figure 1 illustrates how an LLM agent generates the target function for task `scikit-learn_304` from REPOCOD (Liang et al., 2024b), a benchmark for code generation with repository-level dependencies. In the standard setting, the agent is given ① the task description, ② the repository context, and ③ the function specification with docstring. Based on this information, the agent assumes that `y_prob.shape[1]` (the size of distributions) is always greater than 1, since classification should involve at least two classes, and produces the implementation shown in ⑤. However, in `scikit-learn`, developers require functions to also support singleton representation of probability for binary classification (e.g., using `[0.9]` instead of `[0.1, 0.9]` for each distribution). Such a requirement cannot be inferred from ① – ③ alone, and the implementation in ⑤ fails on such inputs. In contrast, the TDD setting additionally provides ④ test cases, which explicitly include usage examples where the binary probability distribution is represented by the singleton second dimension of `y_prob` (the gray lines in ④). This supervision guides the agent to generate the correct implementation in ⑥.

While this highlights TDD as a promising setting, effectively utilizing test cases for code generation is non-trivial. First, a complex function in a real-world repository needs to be validated by dozens of distinct test cases. How to strategically select tests to guide an LLM within a limited context window and computational budget remains a critical open question. Second, related context for function generation is often scattered across the repository, so efficient retrieval is key to guiding accurate implementation. Third, it is crucial to enable the agent to systematically leverage feedback from the test execution to automatically debug and refine its generated code.

To address the above challenges, we propose **TENET**, an LLM agent designed to generate functions within complex repositories under the TDD setting. The agent is built upon three key technical innovations. First, to tackle the challenge of strategically selecting test cases, TENET employs a novel *test harness mechanism (THM)* that utilizes dynamic analysis to select a subset of test cases that invoke target functions from distinct caller functions in the call stack to maximize coverage of diverse target usage scenarios. Second, it leverages a *tailored agent toolset* for repository-level code generation, unifying structural retrieval, semantic similarity search, and interactive debugging, enabling the agent to navigate complex repositories with both breadth and depth. Third, to address the challenge of systematically refining code with test feedback, it implements a *reflection-based refinement workflow (RRW)* that iteratively improves generated code via failure analysis, contextual replenishment, and execution-based debugging. Together, these innovations allow TENET to generate more correct code in complex real-world repositories.

The core contributions can be summarized as follows:

- We propose a novel test harness mechanism for efficient test-driven supervision, which utilizes dynamic analysis to select a small number of test cases that benefit code generation performance.
- We design a tailored agent toolset that supports LLM agents to perform efficient code retrieval with interactive debugging.
- We build a reflection-based refinement workflow for test-aware debugging, enabling LLM agents to more effectively utilize test execution signals and refine incorrect code generations.
- We develop the first test-driven LLM agent for repository-level code generation, namely TENET, which achieves 69.08% and 81.77% Pass@1 on the REPOCOD and RepoEval benchmarks, outperforming the best agentic baselines by 9.49 and 2.17 percentage points (pp), respectively.
- We conduct the first systematic study of test-driven code generation using LLM agents at the repository level. The key findings include:
  - A larger quantity of test cases does not necessarily lead to superior results; we find that a moderate number, typically three to five, often yields optimal performance.
  - Test cases that invoke target functions from distinct callers may provide complementary information and higher test coverage, helping the agent produce the correct implementation.
  - While incorporating test signals in both context retrieval and refinement consistently improves the performance, it comes with higher token comsumptions. We should balance the trade-off between accuracy and efficiency.

## 2 RELATED WORK

### 2.1 TEST-DRIVEN DEVELOPMENT

Test-Driven Development (TDD) is a widely adopted software engineering methodology (Beck, 2022). A typical TDD workflow follows a "test–implement–refactor" cycle: a developer begins with one or more test cases that define a desired function, writes the minimum code necessary to pass those tests, and then refactors the implementation while ensuring all tests continue to pass.

Although prior work shows the effectiveness of TDD (Mathews & Nagappan, 2024; Piya & Sullivan, 2024; Tian et al., 2025), its application to repository-level LLM agents remains underexplored. In this work, we argue that TDD is particularly well-suited for this setting and develop TENET, which achieves state-of-the-art performance on test-driven code generation in real-world repositories.

### 2.2 LARGE LANGUAGE MODEL AGENTS

LLM agents are increasingly used to automate developer workflows, which leverage an LLM as a central reasoning engine to decompose complex problems, create multi-step plans, and interact with environments using a predefined toolset. SWE-Agent (Yang et al., 2024) solves real-world GitHub issues by equipping LLMs with tools for file editing, navigation, and testing. CodeAct (Wang et al., 2024a) is a general agent, utilizes general tools to solve diverse problems. AutoCodeRover (Zhang et al., 2024c) and SpecRover (Ruan et al., 2025) utilize program analysis to perform targeted code modifications. CodeAgent (Zhang et al., 2024a) is the first agent specifically designed for repository-level code generation tasks, relying on a small set of tools and pre-defined workflows.

However, when generating code in a test-driven manner in complex repositories, these agents lack efficient mechanisms for selecting the most beneficial tests from an entire suite and struggle to effectively use execution feedback for code refinement. In contrast, TENET employs the THM to select test cases that provide diverse test scenarios of the target function, yielding a much better context for code generation. Furthermore, TENET enables the LLM to use a more powerful toolset for efficient code retrieval and interactive debugging, and follows the RRW for effective code refinement.

### 2.3 REPOSITORY-LEVEL CODE GENERATION

Repository-level code generation requires models to implement code within complex codebases by reasoning about source code, documentation, and dependencies (Liang et al., 2024b; Zhang et al., 2023; Ding et al., 2023). With the rise of LLMs, many LLM-based approaches have been proposed, which can be categorized into **non-agentic** and **agentic** approaches.

Non-agentic approaches operate without self-planning, but using techniques like Retrieval-Augmented Generation (RAG) and feedback-driven refinement. Some methods use static or learned retrieval strategies (Zhang et al., 2023; Wang et al., 2025; Wu et al., 2024; Shrivastava et al., 2023;

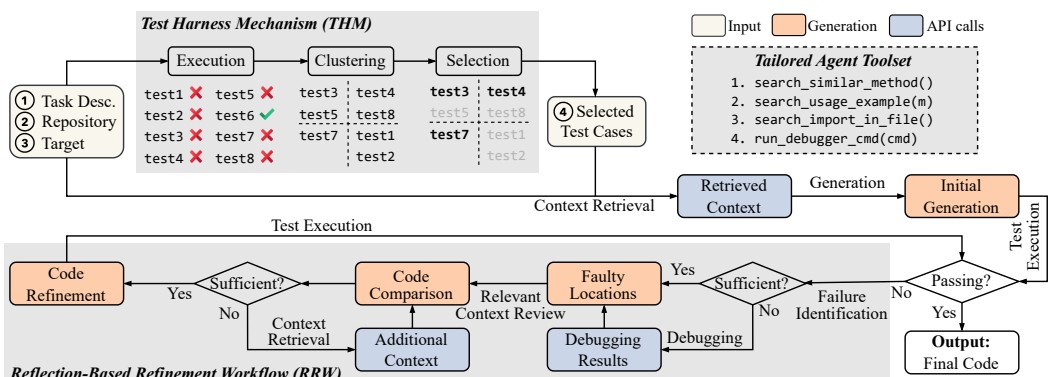

Figure 2: TENET workflow.

Liao et al., 2024), while others base on semantic or structural graphs (Cheng et al., 2024; Liu et al., 2024b; Li et al., 2025; Liang et al., 2024a; Ouyang et al., 2025; Phan et al., 2024). Another category focuses on refinement using external feedback, such as compiler errors (Bi et al., 2024) or symbolic planning (Bairi et al., 2024).

In contrast, agentic approaches remain less explored. CodeAgent equips the LLM with five tools and four fixed planning strategies. Other general-purpose agents like OpenHands, SWE-Agent, and those designed for issue-fixing, AutoCodeRover and SpecRover, while they can be adapted for code generation, they underperform due to their lack of specialized design. In comparison, our proposed agent, TENET, achieves the state-of-the-art accuracy.

## 3 APPROACH: TENET

Figure 2 illustrates the workflow of TENET, consisting of three major components: (1) the *test harness mechnism (THM)*, (2) the *tailored agent toolset*, and (3) the *reflection-based refinement workflow (RRW)*. Specifically, the THM first selects a small set of the most effective test cases to serve as additional input ④ beyond the ① task description, ② repository context, and ③ target function specification, to guide further test-driven code generation. Then TENET explores the repository and retrieves useful context through the *tailored agent toolset* that integrates efficient retrieval with interactive debugging. After collecting sufficient context, TENET makes a generation attempt based on the selected test cases and retrieved context, which is then validated by test execution. If any selected test case fails, TENET enters the RRW to fix the incorrect code.

During the RRW, the agent is required to revisit its fault localization analysis, review the relevant context, and assess whether existing information is sufficient to guide a correct refinement. If the available evidence is not sufficient, the RRW prompts the agent to retrieve additional context and leverage interactive debugging to gather further insights. This reflection loop continues until the agent determines that it has sufficient evidence to perform a fix, at which point it generates a candidate refinement. The refined code is validated through test execution. TENET stays in RRW until a candidate passes all selected test cases, or a maximum limit of refinement attempts is reached.

### 3.1 TEST HARNESS MECHANISM

While TDD is effective in guiding software implementation, dumping the entire test suite to the LLM is infeasible due to two reasons.

- **Effectiveness**. Providing massive test cases can overwhelm the LLM with long, complex prompts, increasing the cognitive load of the model (Mathews & Nagappan, 2024; Liu et al., 2024a).
- **Efficiency**. Such massive test suites and long prompts inevitably incur higher generation latency and execution workload, reducing the efficiency of the agent pipeline (Kim et al., 2024).

To identify a concise and effective subset of test cases, TENET employs a novel *THM* that first executes the full test suite against the unimplemented target function to collect failing cases. These failing cases are then clustered according to the caller function that directly invokes the target function in their call stack. *The intuition is that different caller functions are likely to expose*

*distinct usage patterns and test diverse aspects of the target function's logic, thereby providing complementary test coverage.* For the example shown in Figure 1, test function with call chain `test_log_loss_1_prob_finite` → `log_loss` tests the target function under the binary classification scenario and covers the `if` branch, while another cluster of test functions, with call chain ... → `_backprop` → `log_loss`, all test the target function by covering the `else` branch.

From these clusters, TENET selects at most $T$ test cases. The selection balances two objectives: (1) ensuring diversity by choosing test cases from different clusters, and (2) prioritizing test cases with the shortest call chain from the entry test function to the target function. In practice, the process first attempts to pick one representative test case from each cluster. If the number of clusters is greater than or equal to $T$, the top $T$ clusters are chosen, with one test case selected from each. If the number of clusters is fewer than $T$, additional test cases with the shortest call chains are selected until the budget of $T$ is reached. This strategy ensures that the final subset of test cases remains both diverse across failure patterns and closely tied to the target function's behavior.

We set $T = 3$ for the algorithm described above based on preliminary experiments on the sphinx project from REPOCOD (Section 5.3). We study alternative test selection strategies in 5.4.

## 3.2 TAILORED AGENT TOOLSET

TENET provides a tailored toolset that extends the abstract-syntax-tree (AST)-based toolset of SpecRover (Ruan et al., 2025) for structural context retrieval and interactive debugging. The API toolsets of existing agents can be roughly divided into two categories: AST-based, such as AutoCodeRover and SpecRover, and terminal-command based, such as SWE-Agent (Yang et al., 2024) and OpenHands (Wang et al., 2024b). While AST-based interfaces allow structural navigation, and terminal-command based interfaces offers more flexibility, they still face several limitations when generating functions under repository-level context.

First, RAG techniques can substantially improve generation accuracy by providing relevant code examples. However, existing LLM agents have not incorporated semantic retrieval as an API beyond basic repository navigation, which usually requires multiple attempts to locate the desired context. Second, understanding the use cases is crucial for accurate code generation. Yet, existing LLM agents rely on terminal commands to find the usage of a certain function based on string matching, which is inefficient and error-prone. Third, although interactive debugging plays an important role in refining code (Yuan et al., 2025), existing agents treat it as an end-to-end process, lacking support for fine-grained interactive debugging that enables stepwise evidence collection and fault diagnosis.

To address these, we extend the SpecRover's AST-based toolset with four new APIs:

- **search_import_statement(f)** retrieves all top-level import statements in the specified file `f`. This enables the agent to analyze dependencies and disambiguate call paths in cross-file analysis.
- **search_similar_method(n)** retrieves the top-`n` methods most relevant to the signature and docstring of the target function, ranked by BM25 similarity (Liang et al., 2024b; Zhang et al., 2024a). This API enables the agent to collect context as references efficiently.
- **search_target_usage(n)** retrieves up to n usage examples of the target function via AST analysis. Unlike keyword-matching commands that require multiple indirect queries and often return noisy snippets, this API provides usage contexts in a single step, making it easier for the agent to understand how the target function is invoked.
- **run_debugger_cmd(cmd)** executes a specified debugging command (e.g., `pdb`) within a container session, enabling line-by-line execution such as variable inspection and stack frame traversal for fine-grained debugging (Yuan et al., 2025).

Together, these APIs address the identified limitations by allowing TENET to efficiently retrieve relevant context and conduct structured interactive debugging.

## 3.3 REFLECTION-BASED REFINEMENT WORKFLOW

With the THM and the tailored toolset, TENET first generates a code snippet which is then validated against the chosen test cases. If any test case fails, TENET enters the RRW to revise the code snippet iteratively. Unlike refinement in the setting of self-contained functions, where the LLM only needs to review and revise its own faulty implementation (Chen et al., 2023; Olausson et al., 2023; Huang et al., 2024), the generation of incorrect code within a repository could be due to more complex

reasons. For example, LLM can get "lost in the middle" due to long trajectories (Liu et al., 2024a), overlooking important context such as the usage of repository-specific functions.

To address such issues and control the token consumption, TENET's *RRW* employs an inner loop that requires the LLM to *reflect on its understanding of the faulty implementations*. The LLM is first prompted to identify faulty locations, only invoking debugger commands when necessary to obtain extra signals. Then the LLM reviews the retrieved context to identify relevant snippets (e.g., implementations of similar functionality). When such snippets are available, the LLM is guided to explain their implementations, such as handling of edge cases or the usage of specific functions, and compare them with the faulty implementations to extract insights for bug fixing.

After fault localization and contextual comparison, TENET assesses whether the collected information is sufficient to attempt a refinement. If yes, it formulates and applies a detailed fix strategy. If not, it invokes tools to gather additional context to further examine the behavior of the faulty code. TENET repeats the cycle until enough evidence is gathered to propose a refined solution.

## 4 EXPERIMENTAL SETUP

To assess the effectiveness of TENET and to gain deeper insights into code generation within complex repositories under the TDD setting, we formulate the following research questions (RQs).

- **RQ1:** How effective is TENET compared to other repository-level code generation methods?
- **RQ2:** What are the contributions of TENET's three novel components to the overall performance?
- **RQ3:** What effect does the quantity of test cases in TENET have on code generation performance?
- **RQ4:** How do different test selection strategies affect the code generation pass rate?
- **RQ5:** What is the impact of using test cases at different stages of TENET (e.g., before vs. after initial generation) on performance?

We select two code generation benchmarks with repository-level context, REPOCOD (Liang et al., 2024b) and RepoEval's function-level tasks (Zhang et al., 2023), both of which preserve the full repository content and executable tests, containing 980 and 373 tasks, respectively.

For RQ1, we compare TENET with four strong open-source baselines, covering both non-agentic and agentic approaches. We use Claude Sonnet 4 (Anthropic, 2025) for all baselines and set the temperature to 0. Though different in prompt design, all baselines share the same context as inputs, including the task description, the full repository context, and three randomly selected test cases.

1. **RepoCoder** adopts an iterative retrieval–generation framework (Zhang et al., 2023). We follow the original configuration on RepoEval. For REPOCOD, we use a 12,288-token retrieval window, retrieve up to 30 snippets per query, and cap the maximum completions at 4,096 tokens.
2. **SpecRover** (Ruan et al., 2025) is a multi-agent framework designed for issue resolution task. It coordinates specialized agents for retrieval, generation, testing, and reflection. We follow the same configuration in the paper.
3. **SWE-Agent** is an agent for general software engineering tasks, equipping LLMs with shell commands and custom actions (Yang et al., 2024). The per-task API call limit is set to 50.
4. **OpenHands** is an open-source platform for developing software engineering agents (Wang et al., 2024b). We employ the default CodeAct Agent (Wang et al., 2024a), which supports shell command execution, file reading, and file editing. The per-task API call limit is set to 50.

By default, TENET is set with (1) up to three test cases from the THM, (2) up to 15 retrieval rounds prior to the initial generation, (3) up to five code refinement attempts in RRW, and (4) up to 15 rounds of API calls in the RRW, including both debugging and context retrieval.

For other RQs, we adopt DeepSeek-V3 (DeepSeek-AI et al., 2025) for cost efficiency. To address RQ2, we remove each component from the full TENET system to perform an ablation study on REPOCOD. Removing the THM (TENET$_{-THM}$) provides the agent with full target test suite instead of the selected subset; removing the tailored toolset (TENET$_{-APIs}$) limits the agent to the AutoCodeRover toolset; and removing the RRW (TENET$_{-RRW}$) applies naive refinement given test feedback if test execution fails. To study RQ3, we use the default settings and only change the number of selected tests $T = \{1, 3, 5, 10, All\}$.

For RQ4, we compare TENET's THM with four test selection baselines: three based on test properties and a random selection baseline, detailed below.

| Approaches | REPOCOD | | | RepoEval | | |
|---|---|---|---|---|---|---|
| | Pass@1 (%) ↑ | Avg. Input Cons. ↓ | Avg. Output Cons. ↓ | Pass@1 (%) ↑ | Avg. Input Cons. ↓ | Avg. Output Cons. ↓ |
| RepoCoder | 26.22 | **14,225** | **787** | 53.08 | **6,048** | **306** |
| SpecRover | 33.33 | 95,884 | 6,609 | 72.12 | 52,317 | 4,643 |
| OpenHands | 59.18 | 1,119,149 | 8,988 | 79.60 | 471,103 | 4,434 |
| SWE-Agent | 59.59 | 597,507 | 1,242 | 67.02 | 280,351 | 1,242 |
| TENET | **69.08** | 194,932 | 6,560 | **81.77** | 111,934 | 3,790 |

Table 1: Comparison of Pass@1, LLM token consumptions on REPOCOD and RepoEval.

1. **Random Selection (RS)** serves as a baseline by randomly sampling tests.
2. **Simplicity-Based Selection (SS)** prefers low-cyclomatic-complexity tests (McCabe, 1976).
3. **Failure-Revealing Selection (FRS)** favors tests with explicit assertions or exception checks.
4. **Invocation-Proximity Selection (IPS)** selects tests with shorter call stacks to the target function.

For RQ5, we analyze the impact of using tests at different phases in TENET. In the **NoTest** setting, no tests are provided and the RRW is disabled. In **PreGen**, tests are used only during retrieval before the initial generation. In **PostGen**, tests are applied only in the RRW phase after the initial generation. Finally, in **AllStage**, tests are available throughout the workflow.

## 5 EVALUATION RESULTS

### 5.1 RQ1: BASELINE COMPARE

We report the performance of TENET and other baselines on REPOCOD and RepoEval, including the Pass@1, average input and output token consumptions of the Claude Sonnet 4 in Table 1.

**Performance Superiority.** TENET achieves the highest Pass@1 Chen et al. (2021) across both benchmarks, surpassing the strongest baselines by 9.49 and 2.17 pp respectively. This consistent gain demonstrates the effectiveness of our test-driven design and the great potential of applying the TDD paradigm to code generation with repository-level context.

**Token efficiency**. TENET strikes a strong balance between accuracy and efficiency. RepoCoder, as an non-agentic approach, consumes fewer tokens but yields worse Pass@1. OpenHands and SWE-Agent attain competitive accuracy on REPOCOD, but at the cost of significantly larger input consumption, over 1.12M and 598K tokens, respectively. This overhead stems from: (1) lengthy system prompts that encode detailed execution policies, security constraints, and multi-step workflows, and (2) reliance on terminal-level commands (e.g., `grep`, `find`) that require many sequential steps for code retrieval under the one-command-per-response constraint. These fragmented interactions cause the trajectory to grow rapidly, resulting in a dramatic increase in input token consumption. In contrast, TENET and SpecRover leverage AST-based tools for precise context retrieval and support multiple API calls per response, yielding shorter and denser trajectories. **The results shows that our TENET can achieve both strong accuracy and token efficiency.**

| Variants | Pass@1 (%) ↑ | Avg. Input Cons. ↓ | Avg. Output Cons. ↓ | Avg. API Call ↓ |
|---|---|---|---|---|
| TENET-THM | 31.94 ↓ 17.24% | 208,179 ↑ 40.69 % | 5,547 ↑ 19.41 % | 12.05 ↑ 45.53 % |
| TENET-APIs | 34.29 ↓ 14.89% | 138,031 ↓ 6.72 % | 6,358 ↑ 36.87 % | 10.53 ↑ 27.17 % |
| TENET-RRW | 39.94 ↓ 9.24% | 132,427 ↓ 10.50 % | 4,008 ↓ 13.71 % | 6.62 ↓ 20.05 % |
| TENET | 49.18 | 147,968 | 4,645 | 8.28 |

Table 2: Ablation study of TENET using DeepSeek-V3 on REPOCOD. Compared with the TENET, red arrows indicate worse results and blue arrows indicate better results.

### 5.2 RQ2: CONTRIBUTIONS OF TENET 'S COMPONENTS

Table 2 reports the Pass@1, average token consumptions, and the number of API calls on REPOCOD. Across all cases, removing any component results in a clear decline in Pass@1, as detailed below.

Removing THM causes the largest Pass@1 drop (17.24%). The token consumption also rises substantially with inputs by 40.69% and outputs by 19.41%, along with a 45.53% rise in API calls. This

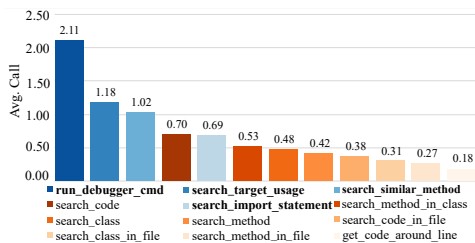

Figure 3: Average API calls per task on RE-POCOD (TENET, DeepSeek-V3).

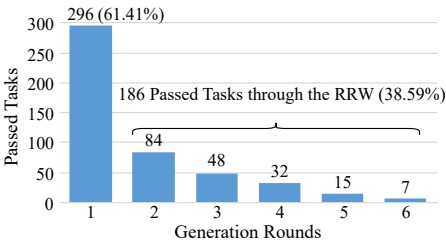

Figure 4: Number of solved tasks per round (TENET, DeepSeek-V3).

| Test Num. | sphinx | seaborn | flask | xarray | sympy | more-itertools | datasets | scikit-learn | astropy | pylint | plotly.py | Total |
|---|---|---|---|---|---|---|---|---|---|---|---|---|
| 1 | 39.39 | 47.44 | 67.44 | 25.30 | 26.80 | 56.98 | 47.46 | 22.61 | 43.52 | 23.08 | **43.42** | 35.71 |
| 3 | **54.55** | 55.13 | 72.09 | **42.17** | **34.02** | 70.93 | **62.71** | **46.18** | 47.06 | 30.77 | 40.79 | **49.18** |
| 5 | 42.42 | 55.13 | **79.07** | 37.35 | 28.87 | **80.23** | 59.32 | 43.95 | **50.59** | **34.62** | 42.11 | 48.57 |
| 10 | 36.36 | **57.69** | 74.42 | 34.94 | 26.80 | 76.74 | 54.23 | 35.35 | 48.24 | **34.62** | 40.79 | 44.29 |
| All | 30.30 | 41.03 | 53.49 | 18.07 | 24.74 | 76.74 | 38.98 | 20.06 | 40.00 | 19.23 | 38.16 | 33.06 |

Table 3: Pass@1 (%) of TENET with different test suite sizes on REPOCOD

is mainly due to two factors. First, each REPOCOD task contains 68 test cases on average[1], far more than the curated suites from the THM. Feeding the model with the full suite introduces redundancy and noise, reducing accuracy. Second, handling the full suite requires analyzing more cases through the pipeline, which increases API calls and token usage. **Overall, these results highlight the necessity of THM in filtering and prioritizing tests, ensuring both efficiency and effectiveness.**

Removing the tailored toolset reduces Pass@1 by 14.89%. Without TENET's specialized tools, such as retrieving semantically similar code, the agent must rely on less efficient APIs to analyze the context, which increases reasoning complexity. Though the input tokens are fewer, the more expensive output tokens and API calls increase substantially. **This shows our tailored toolset improves performance while incurring only a minor increase in input cost, yet substantially reducing output costs and API overhead.**

Removing the RRW leads to a decrease of 9.24% on Pass@1. TENET$_{-RRW}$ naively regenerates code from previously retrieved context and test feedback without explicit reasoning about fault localization or code comparison. As a result, the performance drops, while the token consumption and API calls slightly decrease. **This highlights that RRW improves code generation performance at the cost of increased token usage.**

In addition, Figure 3 shows the average call frequency of different APIs[2], and our four newly introduced APIs (colored in blue) rank among the top five. We observe that `run_debugger_cmd` dominates with the highest frequency (2.11), highlighting the frequent usage of interactive debugging in the RRW. The next most frequently used APIs are `search_target_usage` (1.18) and `search_similar_method` (1.02), suggesting that retrieving usage examples and similar methods is also favorable in the TDD setting. **Together, these observations indicate that the model consistently favors our tailored toolset in the TDD setting.**

Figure 4 shows the number of solved tasks across refinement rounds. Solved tasks drop gradually over rounds, and out of 482 passed tasks in total, 296 (61.41%) are solved in the first attempt. The remaining 186 tasks (38.59%) are recovered through the RRW. **This shows that RRW plays a critical role in rescuing tasks that fail initially, greatly enhancing the effectiveness of TENET.**

### 5.3 RQ3: THE IMPACT OF TEST SUITE SIZE

As a tunable parameter, we set $T = 3$ based on preliminary results on the sphinx project (33 tasks) in REPOCOD, where $T = 3$ achieved the best performance. To address RQ3, Table 3 reports Pass@1 across different test suite sizes on the full REPOCOD benchmark. Overall, $T = 3$ achieves the best results on five projects, covering 59.8% of all tasks (589/980), and also delivers the highest overall

---

[1]This does not distinguish test functions with different inputs. If each input variant is counted as a separate test, the average increases to 313 (Liang et al., 2024b).

[2]The APIs not highlighted in bold are inherited from SpecRover.

| Strategy | Pass@1 (%) ↑ | Avg. Cov (%) ↑ |
|---|---|---|
| RS | 32.86 | 70.94 |
| SS | 36.12 | 72.72 |
| FRS | 38.88 | 71.58 |
| IPS | 41.53 | 76.98 |
| THM | 49.18 | 79.38 |

Table 4: Pass@1 and test coverage under different selection strategies.

Figure 5: Overlap of solved tests across four settings.

Pass@1. $T = 5$ is optimal for four projects (24.5%, 240/980) and ranks second overall. Moreover, performance generally declines as the test suite size increases. **These results show that our finding on sphinx generalizes to the full benchmark. Importantly, more test cases do not necessarily improve outcomes. Instead, a moderate number of tests (three to five) consistently provides reliable gains under the TDD setting.**

### 5.4 RQ4: THE IMPACT OF TEST SELECTION STRATEGIES

Table 4 reports the results of TENET on REPOCOD under different test selection strategies when $T = 3$. Average test coverage is computed based on the ground-truth target functions. All selection strategies outperform the RS baseline, with improvements in Pass@1 generally aligning with higher coverage. Among them, TENET's default THM strategy, which emphasizes caller diversity and invocation proximity, achieves the best results (49.18% Pass@1 and 79.38% coverage). The key takeaway is that **test suites combining caller diversity with invocation proximity provide the most effective guidance for LLM agent code generation with repository-level dependencies.**

### 5.5 RQ5: THE IMPACT OF TEST USAGE STAGE

| Phases | Pass@1 (%) ↑ | Avg. Input Cons. ↓ | Avg. Output Cons. ↓ | Avg. API Calls ↓ |
|---|---|---|---|---|
| NoTest | 29.90 | 32,829 | 2,482 | 5.04 |
| PreGen | 36.93 | 35,427 | 2,408 | 5.63 |
| PostGen | 42.65 | 138,330 | 4,710 | 8.91 |
| AllStage | 49.18 | 147,968 | 4,645 | 8.28 |

Table 5: Results of leveraging tests at different phases in TENET's workflow.

Table 5 reports how leveraging tests at different stages affects the performance of TENET. We make two main observations. First, **leveraging tests in more stages improves correctness.** The Pass@1 rises from 29.90% (NoTest) to 36.93% (PreGen), 42.65% (PostGen), and peaks at 49.18% with AllStage. Second, **this improvement comes at a higher cost**. PostGen and AllStage require more token consumptions and API call counts than NoTest and PreGen, reflecting the extra debugging and context retrieval required by the RRW. Moreover, Figure 5 further analyzes the overlap of solved tasks across the four settings. The AllStage setting achieves the largest number of uniquely solved tasks (66). Each setting also contributes its own distinct set of solved tasks. While some of this diversity may arise from the inherent randomness of LLMs, it also points to potential opportunities for improvement by tailoring how tests are leveraged at different stages.

### 6 CONCLUSION & FUTURE WORK

This work introduces TENET, an agent framework for **repository-level code generation under the TDD paradigm**. It features three components: (1) a test harness mechanism that selects concise and effective tests to guide code generation, (2) a tailored toolset for context retrieval and debugging, and (3) a reflection-based refinement workflow for code fixing. TENET achieves the best performance on two code generation datasets with repository-level context among SOTA baselines. In addition, we present the first study on test suites' impact on code generation with repository-level context, including test numbers, selection strategies, and test usage stage, which offers valuable insights into leveraging TDD for agent-based software development. For future work, we will explore integrating more advanced test generation approaches (Chen et al., 2022; 2024; Schäfer et al., 2024) to overcome the limitation of THM's reliance on existing tests and move toward a fully automated TDD pipeline. We plan to adopt more flexible refinement strategies to further enhance the effectiveness of RRW.

## 7 REPRODUCIBILITY STATEMENT

The implementations of TENET (Section 3) and the evaluation results of the RQs (Section 5) can be downloaded at this link.

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

# A APPENDIX

In the appendix, we provide additional details to complement the main text. First, we include the prompts of TENET and agentic baselines for RQ1 (Section 5.1). Second, we present case studies for RQ2 (Section 5.2), illustrating how each component of TENET contributes to effective code generation, also including a failure case that TENET fails to generate the correct code. Third, we report detailed statistics of the different test selection strategies for RQ4 (Section 5.4) and the complete Pass@1 results on each REPOCOD project for leveraging tests at different stages of TENET (Section 5.5). Finally, we describe the usage of Large Language Models in our paper writing.

## A.1 PROMPT DETAILS OF TENET AND AGENTIC BASELINES

All the following prompts use `scikit-learn_304` from REPOCOD as an example, same as the motivation described in the Section 1.

### A.1.1 TENET

The system prompt and task description example of TENET:

```
You are an intelligent software developer that consistently delivers accurate and reliable
    responses to user instructions. Now you are assigned to a code generation task.
The task description is provided between the tags <issue> and </issue>.
Your goal is to generate an accurate and well-structured implementation for the target
    function.
To do this, you should first iteratively invoke search APIs to retrieve relevant code context
    from the codebase.
Analyze the retrieved context carefully to understand the target functionality, dependencies,
    and any useful patterns or examples that can inform your implementation.
<issue>You are working on a code generation task. You will be provided with:
1. The information of the target function
2. Access to the entire project for retrieval and analysis
3. Tests for the target function (if available)
Your task is to generate the function body of the target.
## Target Code Information:
**Target Function Name:**: 'log_loss';
**File Location:**: 'sklearn/neural_network/_base.py';
**Line Location:**: from line 175 to line 191;
**Source Code:**:
```
def log_loss(y_true, y_prob):
"""Compute Logistic loss for classification.
Parameters
----------
y_true : array-like or label indicator matrix
Ground truth (correct) labels.
y_prob : array-like of float, shape = (n_samples, n_classes)
Predicted probabilities, as returned by a classifier's
predict_proba method.
Returns
-------
loss : float
The degree to which the samples are correctly predicted.
"""
```
## Test Information
We will provide you the top 3 test cases that invoke the target from distinct callers with
    shortest call stack.
Here are 3 selected test cases:
- Test 1:
pytest node id: 'sklearn/neural_network/tests/test_base.py::test_log_loss_1_prob_finite',
    around line: 15.
- Test 2:
pytest node id: 'sklearn/neural_network/tests/test_mlp.py::test_partial_fit_classification',
    around line: 417;
The target function is called in file sklearn/neural_network/_multilayer_perceptron.py around
    line 330;
- Test 3:
pytest node id: 'sklearn/neural_network/tests/test_mlp.py::test_partial_fit_unseen_classes',
    around line: 444;
The target function is called in file sklearn/neural_network/_multilayer_perceptron.py around
    line 330;
## Task Instructions
The target function is currently unimplemented and contains only 'raise NotImplementedError'.
You will have access to different APIs for context retrieval in the codebase.
```

```
Please carefully read the above information and retrieve context wisely to understand the
    target behavior,
and provide a complete solution for the target code.
REMEMBER:
1. Avoid importing additional packages or libraries unless they already exist or considered
    necessary.
2. Ensure your generated code has correct indentation and follows the same formatting style as
    the context.
3. Do not generate additional code or patches other than the above target function.
</issue>
```

The prompt of out tailored agent toolset:

```
Based on the the task, you can use the following search APIs to get more context:
- search_test_cases(): Search for test cases of the target function. Analyzing test cases can
    help you to refine your solution. These test cases are filtered using dynamic analysis
    based on pytest. The API will return the test in pytest nodeid format. Based on the
    pytest nodeid, you can further use other API calls to retrieve the source code of the
    test cases. You don't need to provide any arguments for 'search_test_cases()' API.
- search_import_in_file(file_name: str): Search for top-level import statements in given file
    'file_name'.
- search_target_usage_example(example_num: int): Search for a given number ('example_num') of
    methods that call the target function directly. This will help you to understand how the
    target function is acutally used or tested in the codebase. If 'example_num' is greater
    than the total number of usage examples, the API will return all of them.
- search_test_cases(): Search for test cases of the target function. Analyzing test cases can
    help you to refine your solution. These test cases are filtered using dynamic analysis
    based on pytest. The API will return the test in pytest nodeid format. Based on the
    pytest nodeid, you can further use other API calls to retrieve the source code of the
    test cases. You don't need to provide any arguments for 'search_test_cases()' API.
- search_relevant_method(top_num: int): Search for the method that is most relevant to the
    target function's docstring by default. We will return the 'top_num' methods with the
    highest BM25 score. This may give you hints about the implementation of your target
    function from similar ones.
- run_pdb_cmd(cmd: str): Execute a specified debugging command (e.g., pdb) within a container
    terminal. you can carry out line-by-line execution such as variable inspection and stack
    frame traversal for fine-grained debugging, for example:
    ```
    l --> list source around the current line
    n --> step to the next line (skip into functions)
    s --> step into a function
    c --> continue execution until the next breakpoint
    b 23 --> set a breakpoint at line 23
    p var --> print value of variable var
    q --> quit debugger
    ```
- search_class(class_name: str): search for a class in the codebase. The class signature
    includes class name, base classes, and signatures for all of its methods/properties.
- search_class_in_file(class_name:str, file_name: str): Search for class with name 'class_name
    ' in given file 'file_name'.
- search_method(method_name: str): Search for a method in the entire codebase.
- search_method_in_file(method_name: str, file_path: str): Search for method with name '
    method_name' in file 'file_path'.
- search_method_in_class(method_name: str, class_name: str): Search for method with name '
    method_name' in class with name 'class_name'.
- search_code(code_str: str): Search for a code snippet in the entire codebase. Only 'code_str
    ' is needed.
- search_code_in_file(code_str: str, file_path: str): Search for code snippets conatining '
    code_str' in given file 'file_path'.
- get_code_around_line(file_path: str, line_number: int, window_size: int): Gets the code
    around the specified line_number in the file 'file_path'. 'window_size' is the number of
    lines before and after 'line_number'. Please make sure to provide all 3 parameters.

Remember:

    You MUST provide correct number of arguments when invoking APIs! Do not leave any
        necessary arguments blank.

    You can use multiple APIs in one round.

    Do not call the same API with the same parameters repeatedly.

    You SHOULD NOT generate hallucination code as the API return. We will provide you the
        searched context next round after you providing the needed APIs.

Now analyze the task and select necessary APIs to get more context. It's better to provide the
    APIs you need to call and their arguments in your response.
```

## A.1.2 OPENHANDS

```
You are OpenHands agent, a helpful AI assistant that can interact with a computer to solve
    tasks.

<ROLE>
Your primary role is to assist users by executing commands, modifying code, and solving
    technical problems effectively. You should be thorough, methodical, and prioritize
    quality over speed.
* If the user asks a question, like "why is X happening", don't try to fix the problem. Just
    give an answer to the question.
</ROLE>

<EFFICIENCY>
* Each action you take is somewhat expensive. Wherever possible, combine multiple actions into
     a single action, e.g. combine multiple bash commands into one, using sed and grep to
    edit/view multiple files at once.
* When exploring the codebase, use efficient tools like find, grep, and git commands with
    appropriate filters to minimize unnecessary operations.
</EFFICIENCY>

<FILE_SYSTEM_GUIDELINES>
* When a user provides a file path, do NOT assume it's relative to the current working
    directory. First explore the file system to locate the file before working on it.
* If asked to edit a file, edit the file directly, rather than creating a new file with a
    different filename.
* For global search-and-replace operations, consider using 'sed' instead of opening file
    editors multiple times.
</FILE_SYSTEM_GUIDELINES>

<CODE_QUALITY>
* Write clean, efficient code with minimal comments. Avoid redundancy in comments: Do not
    repeat information that can be easily inferred from the code itself.
* When implementing solutions, focus on making the minimal changes needed to solve the problem
    .
* Before implementing any changes, first thoroughly understand the codebase through
    exploration.
* If you are adding a lot of code to a function or file, consider splitting the function or
    file into smaller pieces when appropriate.
</CODE_QUALITY>

<VERSION_CONTROL>
* When configuring git credentials, use "openhands" as the user.name and "openhands@all-hands.
    dev" as the user.email by default, unless explicitly instructed otherwise.
* Exercise caution with git operations. Do NOT make potentially dangerous changes (e.g.,
    pushing to main, deleting repositories) unless explicitly asked to do so.
* When committing changes, use 'git status' to see all modified files, and stage all files
    necessary for the commit. Use 'git commit -a' whenever possible.
* Do NOT commit files that typically shouldn't go into version control (e.g., node_modules/, .
    env files, build directories, cache files, large binaries) unless explicitly instructed
    by the user.
* If unsure about committing certain files, check for the presence of .gitignore files or ask
    the user for clarification.
</VERSION_CONTROL>

<PULL_REQUESTS>
* When creating pull requests, create only ONE per session/issue unless explicitly instructed
    otherwise.
* When working with an existing PR, update it with new commits rather than creating additional
     PRs for the same issue.
* When updating a PR, preserve the original PR title and purpose, updating description only
    when necessary.
</PULL_REQUESTS>

<PROBLEM_SOLVING_WORKFLOW>
1. EXPLORATION: Thoroughly explore relevant files and understand the context before proposing
    solutions
2. ANALYSIS: Consider multiple approaches and select the most promising one
3. TESTING:
   * For bug fixes: Create tests to verify issues before implementing fixes
   * For new features: Consider test-driven development when appropriate
   * If the repository lacks testing infrastructure and implementing tests would require
       extensive setup, consult with the user before investing time in building testing
       infrastructure
   * If the environment is not set up to run tests, consult with the user first before
       investing time to install all dependencies
4. IMPLEMENTATION: Make focused, minimal changes to address the problem
5. VERIFICATION: If the environment is set up to run tests, test your implementation
    thoroughly, including edge cases. If the environment is not set up to run tests, consult
    with the user first before investing time to run tests.
</PROBLEM_SOLVING_WORKFLOW>
```

```
<SECURITY>
* Only use GITHUB_TOKEN and other credentials in ways the user has explicitly requested and
      would expect.
* Use APIs to work with GitHub or other platforms, unless the user asks otherwise or your task
       requires browsing.
</SECURITY>

<ENVIRONMENT_SETUP>
* When user asks you to run an application, don't stop if the application is not installed.
      Instead, please install the application and run the command again.
* If you encounter missing dependencies:
  1. First, look around in the repository for existing dependency files (requirements.txt,
       pyproject.toml, package.json, Gemfile, etc.)
  2. If dependency files exist, use them to install all dependencies at once (e.g., `pip
       install -r requirements.txt`, `npm install`, etc.)
  3. Only install individual packages directly if no dependency files are found or if only
       specific packages are needed
* Similarly, if you encounter missing dependencies for essential tools requested by the user,
      install them when possible.
</ENVIRONMENT_SETUP>

<TROUBLESHOOTING>
* If you've made repeated attempts to solve a problem but tests still fail or the user reports
       it's still broken:
  1. Step back and reflect on 5-7 different possible sources of the problem
  2. Assess the likelihood of each possible cause
  3. Methodically address the most likely causes, starting with the highest probability
  4. Document your reasoning process
* When you run into any major issue while executing a plan from the user, please don't try to
      directly work around it. Instead, propose a new plan and confirm with the user before
      proceeding.
</TROUBLESHOOTING>

I've uploaded a python code repository in the directory `/testbed`. There is a function
    remained to be completed in `sklearn/neural_network/_base.py`:

<func_signature>
def log_loss(y_true, y_prob)
</func_signature>

File `relevant_test_cases.txt` contains all the test cases that this function need to pass.

Can you help me implement the function described in <func_signature>?

Before your implementation, create a clean working environment for you:
1. Run `git config --global user.email "openhands@all-hands.dev"` and `git config --global
    user.name "OpenHands Bot"` to set your identity.
2. Run `rm -rf .git && git init && git add . && git commit -q -m "init"` to initialize the
    folder as a new Git repo.

Here are the steps for you to follow:
1. Explore the repository to familiarize yourself with its structure.
2. Check the corresponding code of test cases in `relevant_test_cases.txt` to understand the
    expected functionality of the target function.
3. Complete the body of the target function.
4. Execute the test cases in `relevant_test_cases.txt` to ensure your completed function
    passes the test cases.
5. Use the `git diff` command to produce a patch file named `patch.diff` containing your
    implementation changes.

Additional notes:
- When running Python, make sure to use `/opt/miniconda3/envs/testbed/bin/python`.
- Do not change or delete any code that already exists in the repo.
- The `patch.diff` file should be saved in `/workspace`.
```

### A.1.3 SPECROVER

```
You are an intelligent software developer that consistently delivers accurate and reliable
    responses to user instructions. Now you are assigned to a code generation task.
The task description is provided between the tags <issue> and </issue>.
Your goal is to generate an accurate and well-structured implementation for the target
    function.
To do this, you should first iteratively invoke search APIs to retrieve relevant code context
    from the codebase.
Analyze the retrieved context carefully to understand the target functionality, dependencies,
    and any useful patterns or examples that can inform your implementation.
<issue>You are working on a code generation task. You will be provided with:
1. The information of the target function
```

```
2. Access to the entire project for retrieval and analysis
3. Tests for the target function (if available)
Your task is to generate the function body of the target.
## Target Code Information:
**Target Function Name:**: `log_loss`;
**File Location:**: `sklearn/neural_network/_base.py`;
**Line Location:**: from line 175 to line 191;
**Source Code:**:
```
def log_loss(y_true, y_prob):
"""Compute Logistic loss for classification.
Parameters
----------
y_true : array-like or label indicator matrix
Ground truth (correct) labels.
y_prob : array-like of float, shape = (n_samples, n_classes)
Predicted probabilities, as returned by a classifier's
predict_proba method.
Returns
-------
loss : float
The degree to which the samples are correctly predicted.
"""
```
## Test Information
```
sklearn/neural_network/tests/test_mlp.py::test_partial_fit_classification
sklearn/tests/test_common.py::test_estimators[MLPClassifier(max_iter=100)-
    check_f_contiguous_array_estimator]
sklearn/neural_network/tests/test_mlp.py::test_gradient
```
## Task Instructions
The target function is currently unimplemented and contains only `raise NotImplementedError`.
You will have access to different APIs for context retrieval in the codebase.
Please carefully read the above information and retrieve context wisely to understand the
    target behavior,
and provide a complete solution for the target code.
REMEMBER:
1. Avoid importing additional packages or libraries unless they already exist or considered
    necessary.
2. Ensure your generated code has correct indentation and follows the same formatting style as
    the context.
3. Do not generate additional code or patches other than the above target function.
</issue>
```

## A.1.4   SWE-AGENT

```
SETTING: You are a helpful assistant and a senior developper that can interact with a computer
    terminal and other provided tools to solve code generation tasks.

The special interface consists of a file editor that shows you {{WINDOW}} lines of a file at a
    time.
In addition to typical bash commands, you can also use the following commands to help you
    navigate and edit files.

COMMANDS:
{{command_docs}}

Please note that THE EDIT COMMAND REQUIRES PROPER INDENTATION.
If you'd like to add the line ' print(x)' you must fully write that out, with all those spaces
    before the code! Indentation is important and code that is not indented correctly will
    fail and require fixing before it can be run.

RESPONSE FORMAT:
Your shell prompt is formatted as follows:
(Open file: <path>) <cwd> $

You need to format your output using two fields; discussion and command.
Your output should always include _one_ discussion and _one_ command field EXACTLY as in the
    following example:
DISCUSSION
First I'll start by using ls to see what files are in the current directory. Then maybe we can
    look at some relevant files to see what they look like.
```
ls -a
```
```

```
You should only include a *SINGLE* command in the command section and then wait for a response
    from the shell before continuing with more discussion and commands. Everything you
    include in the DISCUSSION section will be saved for future reference.
If you'd like to issue two commands at once, PLEASE DO NOT DO THAT! Please instead first
    submit just the first command, and then after receiving a response you'll be able to
    issue the second command.
You're free to use any other bash commands you want (e.g. find, grep, cat, ls, cd) in addition
    to the special commands listed above.
However, the environment does NOT support interactive session commands (e.g. python, vim), so
    please do not invoke them.

We're currently attempting to solve the following code generation problem:
ISSUE:
I've uploaded a python code repository in the directory '/testbed'. There is a function
    remained to be completed in 'sklearn/neural_network/_base.py':

<func_signature>
def log_loss(y_true, y_prob)
</func_signature>

The test cases that this function need to pass are:

<tests>
sklearn/neural_network/tests/test_mlp.py::test_partial_fit_unseen_classes
sklearn/neural_network/tests/test_mlp.py::test_lbfgs_classification[X0-y0]
sklearn/neural_network/tests/test_mlp.py::test_mlp_warm_start_with_early_stopping[
    MLPClassifier]
</tests>

Please only use the above tests to verify your implementation. Do not use other tests or write
    your own.

Please help me implement the function described above.

Now, you will start solve this issue on your own. Your terminal session has started and you're
    in the repository's root directory. You can use any bash commands or the special
    interface to help you. **Edit only the function to be completed and run tests you want.**
When you're satisfied with all of the changes you've made, you can submit your changes to the
    code base by simply running the submit command.
Note you cannot use any interactive session commands (e.g. python, vim) in this environment,
    but you can use '/opt/miniconda3/envs/testbed/bin/python' to run a python file.
NOTE ABOUT THE EDIT COMMAND: Indentation really matters! When editing a file, make sure to
    insert appropriate indentation before each line!

INSTRUCTIONS:
0. Run 'rm -rf .git && git init && git add . && git commit -q -m "init"' first to initialize
    the folder as a new Git repo. THIS IS A MUST!
1. Quickly find the file where the target function is located and find the specific line
    number where the target function is located.
2. Explore the repository and collect necessary context to familiarize yourself with the repo
    and the target.
3. Utilize the test cases to make sure the completed function passes the test cases. DO NOT
    USE EXTRA TESTS OR WRITE YOUR OWN.
4. when you believe you finish the task, use the 'submit' action to submit the task.

TIPS:
0. ONLY complete the body of the target function, and DO NOT change or delete any code that
    already exists in the repo.
1. If you open a file and need to get to an area around a specific line, using the goto
    command, such as 'goto 583', is much quicker.
2. Make sure to look at the currently open file and the current working directory (which
    appears right after the currently open file). The currently open file might be in a
    different directory than the working directory! Note that some commands, such as 'create
    ', open files, so they might change the current open file.
3. When editing files, it is easy to accidentally specify a wrong line number or to write code
    with incorrect indentation. Always check the code after you issue an edit to make sure
    that it reflects what you wanted to accomplish. If it didn't, issue another command to
    fix it.
4. MKAE SURE your ouput in each round only consider ONE discussion and ONE command! Please
    wait for a response from the shell before continuing with more discussion and commands.
5. Again, if all three provided tests passes, you no longger need do extra tests. You may
    consider yourself already finished the task.
Now, let's start solving the task.

(Open file: )
(Current directory: /testbed)
bash-$
```

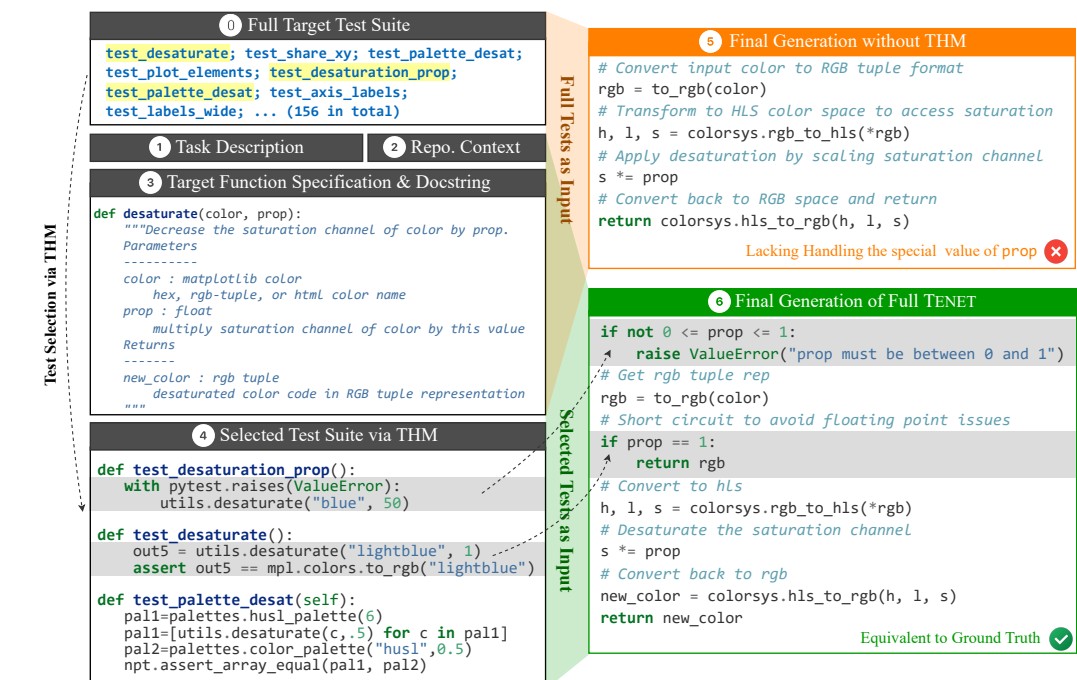

Figure 6: A case study on task `seaborn_34` from REPOCOD. This example explains how the test cases selected by the THM guides the TENET toward correct code generation.

## A.2 CASE STUDIES

In this part, we provide case studies about how the test harness mechanism (THM), tailored agent toolset, and the reflection-based refinement workflow (RRW) contribute to the code generation performance. We also provide a failure case study to indicate possible limitations of our TENET.

### A.2.1 TEST HARNESS MECHANISM

Figure 6 demonstrates the task `seaborn_34` from REPOCOD to show the effectiveness of the THM. The full target test suite of `seaborn_34` contains 156 test cases for the target function `desaturate`. Without the THM, The ⓪ test suite with massive size confuses the LLM, causing the agent ignoring important test signals and finally generating the incorrect code in ⑤. First, it did not follow the input validation requirements, which is explicitly conveyed in `test_desaturation_prop`. Second, it fails the test `test_desaturate` with the following error message.

```
E AssertionError: assert (0.6784313725...9607843137256) == (0.6784313725...9607843137255)
E At index 1 diff: 0.8470588235294119 != 0.8470588235294118
```

The failure occurs since the incorrect code in ⑤ always performs an RGB→HLS→RGB round-trip, even for the boundary case `prop==1`. This introduces tiny floating-point deviations and causes the test to fail. In contrast, the THM selects three test cases without overwhelming the agent, and finally the TENET uses two rounds of refinement to generate the correct code in ⑥ that carefully handles all the edge cases.

### A.2.2 TAILORED AGENT TOOLSET

Figure 7 shows the task `scikit_47` from REPOCOD, explaining how the tailored agent toolset improves the efficiency and generation accuracy of the TENET. Without our tailored toolset, the LLM first searches for the class `FeatureUnion` in the file `sklearn/pipeline.py`: `search_class_in_file("FeatureUnion", "sklearn/pipeline.py")`, which is the class to which the target function `fit` belongs. Then LLM retrieves the target function specifications within the class `FeatureUnion`: `search_method_in_class("fit", "FeatureUnion")`, though they are already provided as the input ③. After these two API calls, the agent believes the collected

context is sufficient for a generation attempt. However, the initial generation fails all three test cases with similar error message.

```
sklearn/tests/test_pipeline.py:508: in test_feature_union
    fs.fit(X, y)
sklearn/pipeline.py:1653: in fit
    results = Parallel(n_jobs=self.n_jobs)(
sklearn/utils/parallel.py:77: in __call__
    return super().__call__(iterable_with_config)
/usr/local/lib/python3.10/site-packages/joblib/parallel.py:1918: in __call__
    return output if self.return_generator else list(output)
/usr/local/lib/python3.10/site-packages/joblib/parallel.py:1847: in _get_sequential_output
    res = func(*args, **kwargs)
sklearn/utils/parallel.py:139: in __call__
    return self.function(*args, **kwargs)
sklearn/pipeline.py:1298: in _fit_one
    return transformer.fit(X, y, **params["fit"])
E TypeError: 'NoneType' object is not subscriptable
```

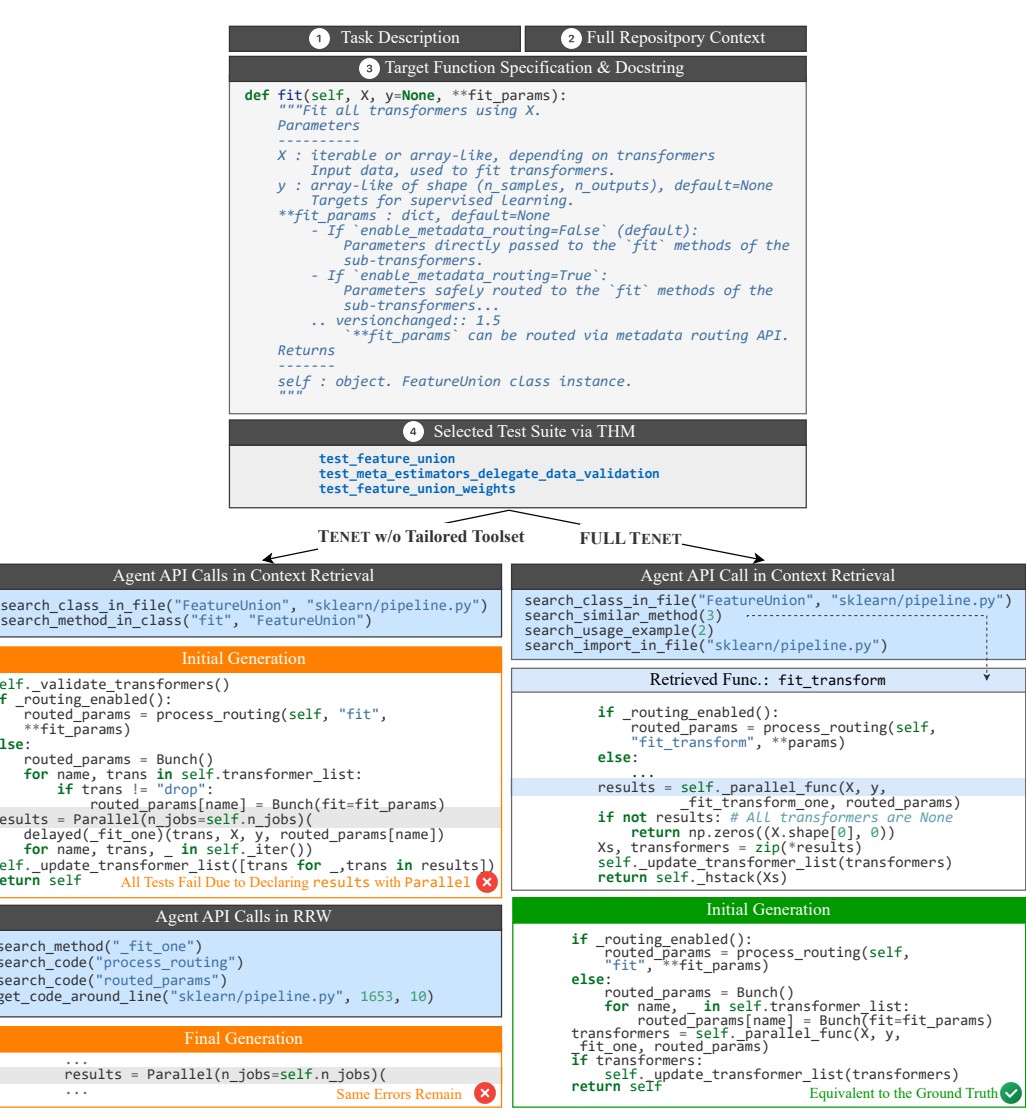

Figure 7: A case study on task `scikit_47` from REPOCOD. This example explains how the tailored agent toolset contributes to the efficiency and effectiveness of the TENET.

The error persists because the agent repeatedly assumes that the target function `fit` should use the `Parallel` class. This misconception arises from the frequent use of `Parallel` for parallel execution within the `FeatureUnion` class. Even after entering the RRW and calling other APIs to

retrieve context based on failure feedback, the agent continues to misapply `Parallel`, leaving the error unresolved in the end.

In contrast, with the support of our tailored agent toolset, Except from retrieving the context of the class `FeatureUnion`, TENET also uses `search_similar_method(3)`, `search_usage_example(2)`, and `search_import_in_file("sklearn/pipeline.py")` to search for similar methods in the codebase , usage examples of the target `fit` function, and the imports avaliable in the file . Among them, `search_relevant_method(3)` returns the context of function `fit_transform` that uses `_parallel_func` for parallel execution. TENET catches this insight and generates the correct implementation without any refinements. This case reveals that our tailored toolset can efficiently help the agent retrieve useful context and improve the generation accuracy.

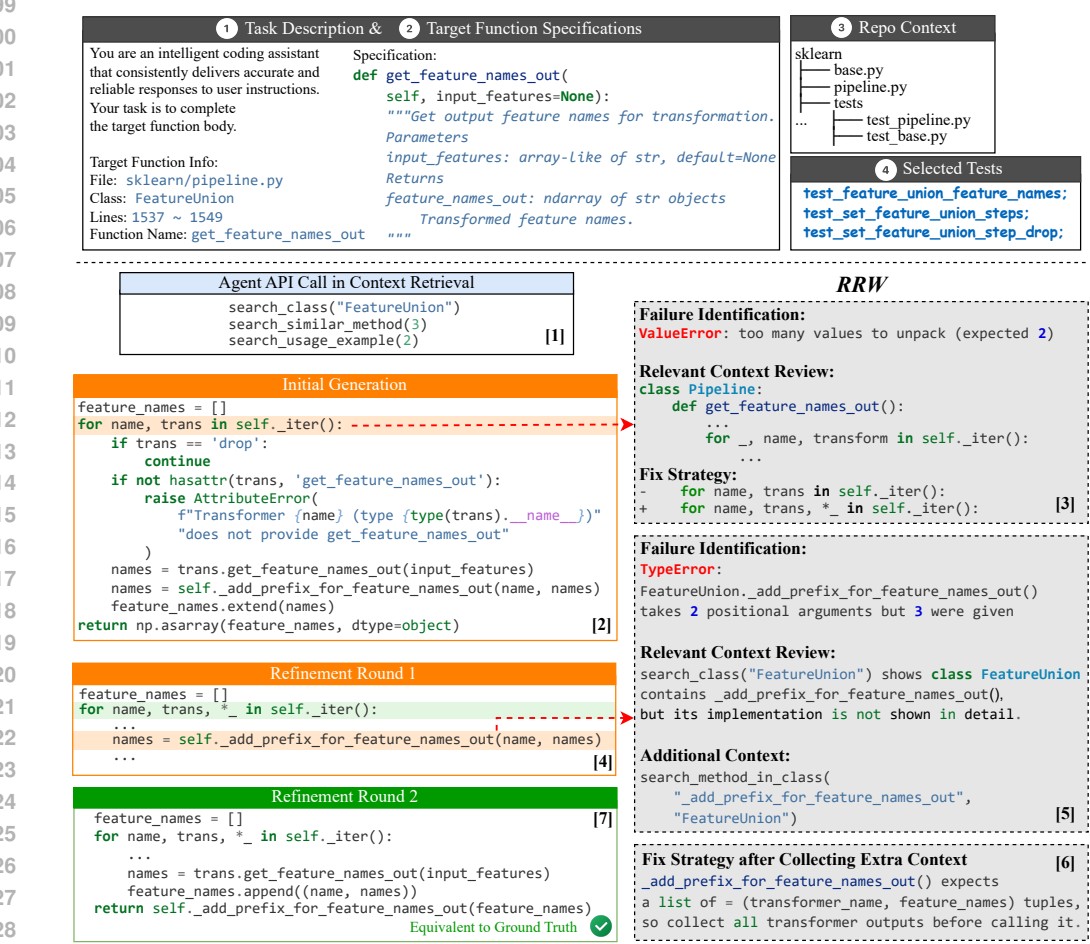

Figure 8: A case study on task `scikit_49` from REPOCOD. This example explains how the RRW contributes to the efficiency and effectiveness of the code refinement in TENET.

### A.2.3 REFLECTION-BASED REFINEMENT WORKFLOW

Figure 8 illustrates the task `scikit_49` from REPOCOD to show how the RRW enables TENET to progressively correct generation errors and converge to the ground-truth solution.. Given the ① task description, ② the specification of the target function `get_feature_names_out`, ③ full repository context, and the ④ selected test cases with source code, TENET first calls three APIs in [1] to retrieve the context of the class `featureUnion` to which the target function belongs, similar methods and the usage examples of the target. Believing the context sufficient, TENET generates an initial implementation [2]. However, test execution immediately exposes a runtime error in [3]: *ValueError: too many values to unpack*. Guided by RRW, TENET first identifies the failure location (the `for` loop),

then reviews the retrieved context. Finding the similar loop in function `get_feature_names_out`, TENET corrects the loop structure and produces the first refinement in [4].

When re-tested, the code triggers a new error: *TypeError: _add_prefix_for_feature_names_out takes 2 positional arguments but 3 were given.* Based on the clear execution feedback, the RRW guides TENET to identify the fauilure location quickly and find that the skeleton of `_add_prefix_for_feature_names_out` has been retrieved but its full implementation is missing by context review. To resolve this, TENET issues an additional query `search_method_in_class("_add_prefix_for_feature_names_out", "FeatureUnion")` to obtain the complete method definition, shown in [5]. With the correct usage clarified, TENET applies the appropriate fix strategy in [6] and produces a final refinement [7] that is equivalent to the ground truth.

By iteratively identifying failure signals, reviewing context, and adaptively invoking APIs when necessary, the RRW is able to help agents overcome non-trivial generation errors and improve accuracy.

### A.2.4 FAILURE CASE STUDY

In Figure 9, We use task `more_itertools-66` from REPOCOD as a failure case stud. The TENET is asked to generate the target function `windowed` at file `more_itertools/more.py` from line 870 to 896. The agent first invokes three APIs in [1] to search for the similar method, usage examples of the target, and the top-level import statements at file `more_itertools/more.py`. After analyzing the context, TENET reasons that the target function `windowed` should generate sliding windows, support padding for incomplete windows and custom step sizes, and return results consistent with tests. The function should involve input validation, iterator handling, first-window construction, sliding logic, and step control. The first attempt in [2] passes the first and the second tests but fails on the third one with the error message: *UnboundLocalError: cannot access local variable `consume` where it is not associated with a value.* Since the test feedback is easy to trace, the agent identifies the failure location immediately, and found the function `consume` is already imported in the file, which does not need to self-define in the branch `if step > 1`.

After analysis, TENET generates the second version of implementation in [4]. However, the third test fails again, indicating the logic mistake in generating the sliding windows. This is where TENET starts to hallucinate. After context review in [5], TENET indicates that the collected information demonstrates using function `tee` and `zip_longest` can fix the logic error. However, The two retrieved context window in the figure show that `tee` function is only used once through the entire collected context, and `zip_longest` function is never used after import statements. The hallucination causes TENET generate the second refined code in [6] but fails to pass the third test.

then in the rest of the workflow, TENET insists using function `tee` and `zip_longest`, leading to failure in the end. This case highlights a limitation of TENET: when context signals are weak or misleading, the agent may overfit to spurious cues, hallucinate dependencies, and persist in unproductive refinements, preventing the correct generation.

| Strategy | seaborn | flask | xarray | sphinx | sympy | more-itertools | datasets | scikit-learn | astropy | pylint | plotly.py |
|----------|---------|-------|--------|--------|-------|----------------|----------|--------------|---------|--------|-----------|
| RS | 44.87 | 76.74 | 32.53 | 42.42 | 28.87 | 68.60 | 44.07 | 12.10 | 34.12 | 19.23 | 36.84 |
| SS | 51.28 | 69.77 | 36.14 | 45.45 | 32.99 | 68.60 | 47.46 | 15.97 | 36.47 | 23.08 | 43.42 |
| FRS | 47.44 | 74.72 | 30.12 | 36.36 | 32.99 | 63.95 | 50.84 | 28.98 | 36.47 | 19.23 | 40.79 |
| IPS | 55.13 | 74.42 | 33.73 | 48.48 | 29.90 | 69.77 | 50.84 | 28.66 | 47.06 | 26.92 | 42.11 |
| THM | 55.13 | 72.09 | 42.17 | 54.55 | 34.02 | 70.93 | 62.71 | 46.18 | 47.06 | 30.77 | 40.79 |

Table 6: Pass@1 of TENET (DeepSeek-V3) under different selection strategies across different repositories from REPOCOD.

### A.3 STATISTICS OF DIFFERENT TEST SELECTION STRATEGIES

This section we will provide more details about the test selection strategies mentioned in Section 5.4. Table 6 shows the complete Pass@1 of TENET based on DeepSeek-V3 across each project on RE-POCOD. Our THM outperforms consistently than other selection baselines, further demonstrating the effectiveness of our design of the THM.

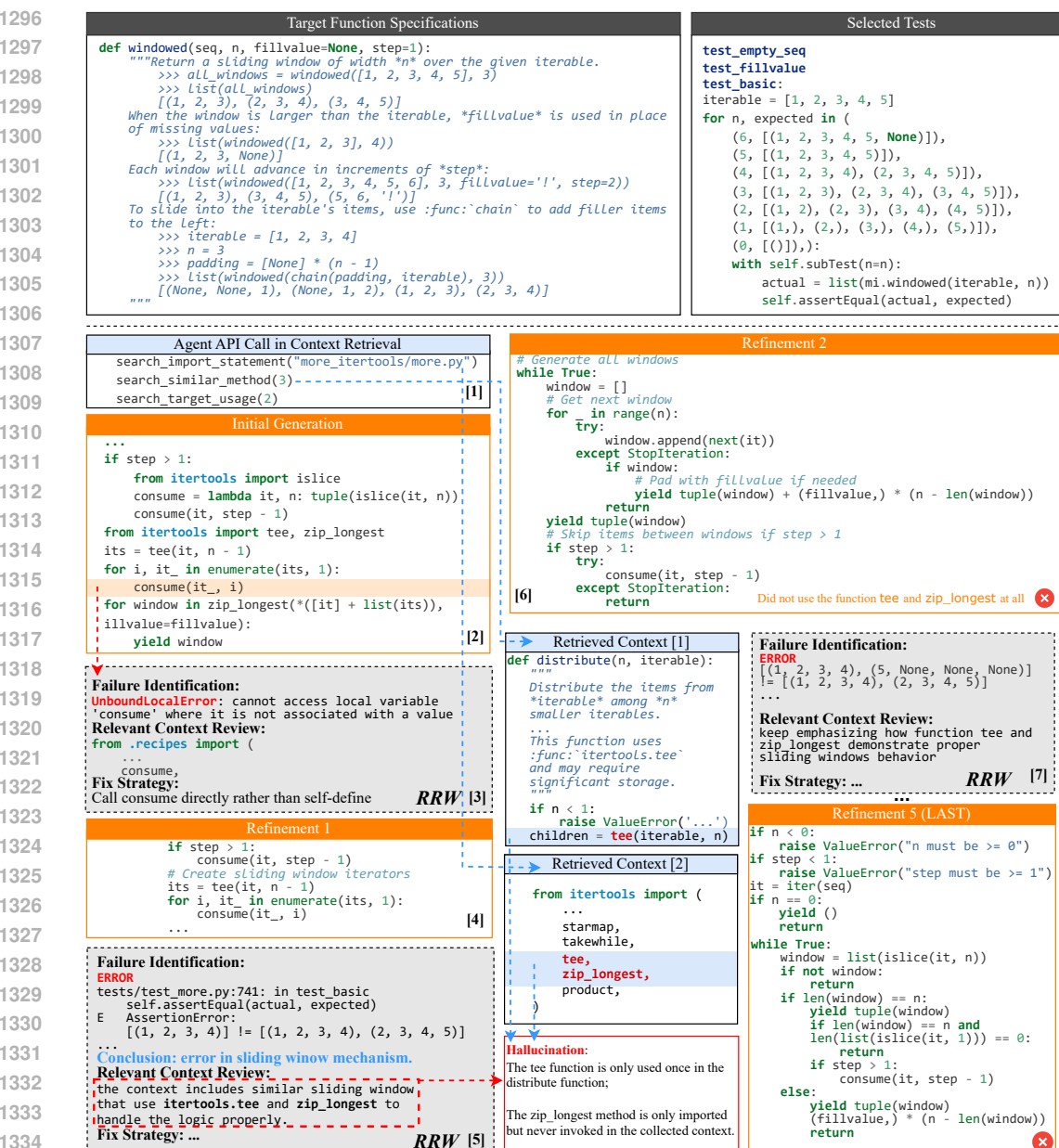

Figure 9: A failure case study on task `more_itertools-66` from REPOCOD.

### A.3.1 RANDOM SELECTION

Random Selection (RS) acts baseline that uniformly samples a fixed number $T$ of test cases from the full test suite. It helps benchmark the impact of smarter selection strategies.

### A.3.2 SIMPLICITY-BASED SELECTION

Simplicity-Based Selection (SS) prioritizes test cases with lower cyclomatic complexity, under our assumption that simpler tests are more likely to isolate specific behaviors and yield focused feedback.

Test cases vary in complexity. Some are concise and directly validate specific behavior, while others with multiple branches and complex logics. We hypothesize that simpler tests with fewer control-flow paths are more effective for validation and refinement, as they contain limited number of control paths or dependencies. In contrast, complex test functions often involve auxiliary logic, nested calls,

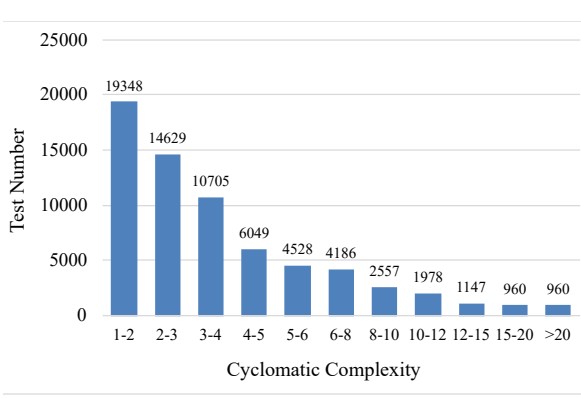

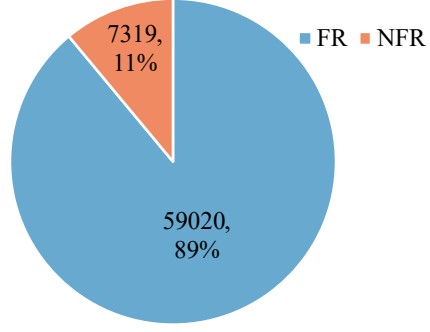

Figure 10: Test distributions on REPOCOD based on cyclomatic complexity.

Figure 11: Test distributions on RE-POCOD based on FRS strategy. FR: tests containing failure revealing structure; NFR: tests that have no failure revealing structure.

or multiple assertions, making their failures harder to interpret and potentially misleading for the agent.

To quantify test simplicity, we compute the cyclomatic complexity of each test function and sort them in ascending order. Then we select the top-k tests as the final test set according to the given test number $T$. Figure 10 presents the distribution of test cases on REPOCOD with respect to cyclomatic complexity. Most tests fall within a complexity range of one to four, and their frequency decreases as complexity increases. Although each task on REPOCOD is accompanied by a relatively large full test suite (on average 68 tests per task), more than 45 tests per task have a complexity below four. This indicates that the majority of existing tests are of relatively low complexity, suggesting that simple and clear tests remain more practical and useful for both LLMs and developers.

### A.3.3 FAILURE-REVEALING SELECTION

Failure-Revealing Selection (FRS) prefers tests that contain explicit assertions or exception checks.

Not all test cases may produce direct and clear signals. Tests that simply run code without checking behavior (e.g., smoke tests or loose integration tests) often pass silently or fail ambiguously. In contrast, tests that include assertions, such as assert and raise, are likely written to verify specific properties. When these tests fail, they typically produce direct and interpretable signals, such as mismatched values or unhandled exceptions. By selecting such tests, their source code and resulting signals may provide the agent with clearer guidance on specific behaviors. We include a test function in the Failure-Revealing Selection when abstract syntax tree (AST) parsing reveals the presence of any of the following constructs.

- Python built-in `assert` or `raise` statements.
- Pytest constructs such as `with pytest.raises()`.
- Unittest-style assertions including `self.assert*` methods (e.g., `self.assertEqual`, `self.fail()`) and their variants.

From all qualified tests, we randomly sample $T$ test cases to be provided to the agent. Figure 11 shows the test distribution of REPOCOD under the FRS strategy. Nearly 90% of the test cases in REPOCOD include constructs such as codeassert or `raise`. This suggests that in real-world development scenarios, developers indeed frequently encounter tests containing such assertions, highlighting the practicality and realism of our strategy design. However, the drawback is that the selection pool becomes overly large, covering almost 90% of all test cases. This may cause the results to be similar to RS and thereby diminish the effectiveness of this strategy.

### A.3.4 INVOCATION-PROXIMITY SELECTION

Invocation-Proximity Selection (IPS) prioritizes test cases with shorter call chains to the target function, based on the assumption that more direct invocations yield clearer execution feedback.

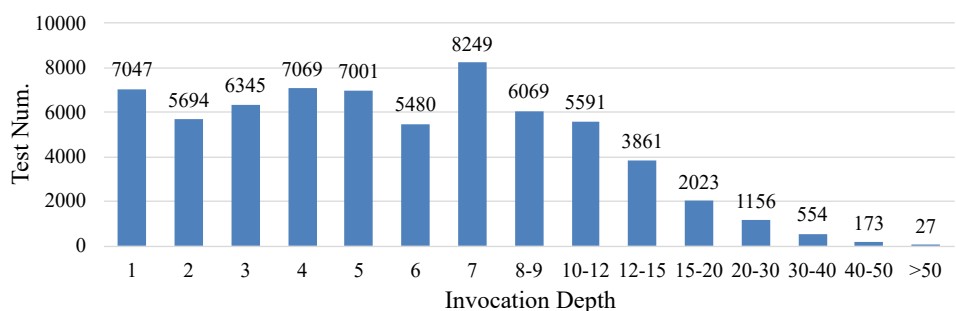

Figure 12: Test distributions on REPOCOD based on the invocation depth from the test function to the target function.

| Stage | seaborn | flask | xarray | sphinx | sympy | more-itertools | datasets | scikit-learn | astropy | pylint | plotly.py |
|---|---|---|---|---|---|---|---|---|---|---|---|
| NoTest | 41.03 | 53.49 | 18.07 | 30.30 | 24.74 | 51.16 | 38.98 | 20.06 | 37.65 | 19.23 | 28.95 |
| PreGen | 41.03 | 60.46 | 19.28 | 27.27 | 18.56 | 61.63 | 44.07 | 36.94 | 41.18 | 19.23 | 34.21 |
| PostGen | 51.28 | 72.09 | 31.33 | 36.36 | 27.84 | 67.44 | 54.23 | 40.76 | 35.29 | 26.92 | 46.05 |
| AllStage | 55.13 | 72.09 | 42.17 | 54.55 | 34.02 | 70.93 | 62.71 | 46.18 | 47.06 | 30.77 | 40.79 |

Table 7: of leveraging tests at different phases in TENET's workflow across different repositories in REPOCOD.

The clarity of test context plays a critical role in improving the agent's understanding of the target functionality. When a test case invokes the target function through a shorter call chain, the function context and the execution signal are often more concise and relevant to the target function. In contrast, tests that reach the target through multiple layers of indirection often introduce additional abstractions and complex logic, which dilute useful signals and obscure the root cause of failures. By favoring tests with shorter call chains, we aim to provide the agent with cleaner behavioral signals and thereby strengthen its reasoning and refinement process.

To measure the invocation depth from tests to targets, we replace the body of the target function with `raise NotImplementedError` and execute the full target test suite using `pytest`. For each test case, we extract the call chain from the resulting traceback and record the depth at which the target is invoked. Test cases are then ranked by this depth in ascending order, and the top-k test cases are selected for use.

Figure 12 reports the test distribution on REPOCOD based on the invocation depth from the test to the target. The majority of tests concentrate at shallow depths. invocation depth from one to five each has over 5,000 tests, with a significant peak at depth seven (8,249 tests). Then as the invocation depth increases, the number of tests steadily drops. This distribution suggests that in practical development, developers tend to provide tests that directly or closely invoke the target function, which is easier to trace and debug through validation.

## B COMPLETE PASS@1 OF LEVERAGING TESTS AT DIFFERENT STAGES

In Section 5.5, we present the main results and the overlap of solved tasks when utilizing test cases across different stages of TENET. Here we further report the complete Pass@1 for each repository in REPOCOD, as shown in Table 7. AllStage achieves the best performance on 10 projects except the plotly.py. This highlights the overall effectiveness of incorporating tests throughout all stages of generation and the potential of the TDD setting in modern software development with LLMs.

## C LLM USAGE

We use LLMs solely to polish the manuscript. Their roles were limited to checking grammar, improving readability, and ensuring clarity of expression. No substantive changes to the content, analysis, or results were made using the LLM.

