# OpenReview forum: "TENET: Leveraging Tests Beyond Validation for Code Generation"
_ICLR.cc/2026/Conference — ICLR 2026 Conference Withdrawn Submission_

### Official Review · Reviewer_hczU · 2025-10-27

**Soundness:** 3
**Presentation:** 3
**Contribution:** 2
**Rating:** 2
**Confidence:** 4

**Summary:**

The paper proposes **TENET**, an LLM-based agent framework for repository-level code generation under a *test-driven development (TDD)* setting. It introduces three key components:

1. **Test Harness Mechanism (THM)** — selects a small and diverse subset of test cases via dynamic analysis of call stacks;
2. **Tailored Agent Toolset** — adds APIs for semantic retrieval, usage example extraction, and interactive debugging;
3. **Reflection-based Refinement Workflow (RRW)** — iteratively analyzes failed executions and refines code using retrieved context.

Experiments on **REPOCOD** and **RepoEval** show that TENET achieves the best Pass@1 results (+9.49% and +2.17% over the strongest baselines) while reducing token consumption compared to SWE-Agent and OpenHands. The paper also conducts ablations on test number, selection strategies, and usage stages, claiming that TDD signals can significantly improve repository-level code generation.

**Strengths:**

- **Comprehensive empirical study.**

  The paper conducts well-organized experiments (five RQs, multiple baselines, extensive ablations) with detailed reporting of efficiency, token cost, and API calls. The experimental section is clearly written and reproducible.

- **Solid engineering implementation.**

  The system integrates retrieval, debugging, and refinement within a single agent framework, showing thoughtful design and clean modularity. The new APIs and ablation results demonstrate real engineering effort.

- **Clear presentation.**

  Figures and workflow descriptions are well structured. The writing is smooth and polished, making it easy to follow even for readers outside the TDD or LLM-agent subdomain.

**Weaknesses:**

- **Conceptual mismatch between TDD and LLM code generation.**

  The entire premise—that future LLM-based development will follow TDD discipline—is idealized and unrealistic. In practice, developers rarely write tests first, and when LLMs are used, both code and tests are often co-generated (“vibe coding”). Thus, tests here act as **evaluation oracles**, not genuine specifications. The “TDD setting” effectively reduces to *test-augmented prompting* rather than a new development paradigm.

- **Limited generality and ecological validity.**

  All experiments are restricted to **single-function completion** within Python repositories. Real-world agentic coding (e.g., SWE-Bench, multi-file issue fixing) involves multi-step planning, environment setup, dependency management, and partial refactoring—none of which TENET handles. Therefore, the practical relevance of the results to real software engineering workflows is doubtful.

- **Over-claiming conceptual novelty.**

  The three components (test selection, retrieval, iterative refinement) correspond closely to existing ideas:

  - THM ≈ dynamic test prioritization;
  - Tailored Toolset ≈ retrieval APIs in SpecRover / SWE-Agent;
  - RRW ≈ self-debug / reflection loops.

  The paper repackages known mechanisms under TDD terminology without providing new theoretical or behavioral insights.

**Questions:**

N/A

---

### Official Review · Reviewer_mgCD · 2025-10-28

**Soundness:** 2
**Presentation:** 3
**Contribution:** 2
**Rating:** 2
**Confidence:** 5

**Summary:**

This work present Tenet, a method for leveraging test cases for code generation on repository level tasks. Concretely, the method contains a specialized method to choose specific relevant tests for a function to implement and provides them to an LLM prior to tasking it with the implementation. During implementation, the method used AutoCodeRover with additional tooling to search the codebase and run debugger commands. After implementation, the tests are run against the implementation, and the LLM is tasked with refining its solution in case of test failure. They evaluate their method against other methods based on SWE-Agent and OpenHands and demonstrate higher scores on RepoEval and RepoCod benchmarks. In ablations, they show that removing components from the pipeline, or replacing them with alternative implementations, as well as changing the number of relevant tests provided to the agent generally reduces performance.

**Strengths:**

Investigating which tests are most helpful to guide function implementation is an interesting idea and deserves further research.
The key finding of this method is that providing only specific test cases to an LLM to guide function implementation aids code generation, which provides interesting insights in this direction. The most improtant results for this are evident from Tables 3, 4 and 5, which ablate the test selection strategies and amounts used by Tenet.
The authors further propose specialized tools to search the code base and debug code, and execution feedback from tests, which are reasonable additions to code generation setups.
The figures and tables are legible.

**Weaknesses:**

It is not very clear to me that Tenet indeed outperforms simple agentic setups, and it is not very clear how much exactly each of the proposed improvements contributes to improved performance over SOTA agents. This is due to fundamentally incomparable setup of related work:
 - Tenet appears to run test cases automatically for the agent. This is opposed to the standard setup of SWE-agent and OpenHands, which need to figure out how to run test cases and then correctly invoke them during execution. This also leads naturally to more API calls.  The SWT-bench authors [1] found that LLM-based agents often struggle exactly there. A fair comparison would similarly run the test suite for SWE-agent and OpenHands, but for example not pick the relevant test cases for them.
 - Tenet picks the most relevant test cases based on a sophisticated heuristic. The authors note that SWE-Agent and Openhand are provided with three _random_ test cases (L.301). According to the OpenHands prompt (A.1.2), the test case results are then only provided in a text file in the repository, which requires an API call to read. The SWE-Agent is even instructed to only focus on the provided test cases (A.1.3) which, as pointed out, are random. This means that competing methods may be adversely configured, resulting in lower performance. It is not clear if standard agents would, e.g., be able to figure out test cases relevant to their implementation on their own without these modifications.
 - The authors provide specialized tools to the agent to search the codebase. Without these tools, the agent performs significantly worse (Table 2). It is unclear if providing OpenHands and SWE-Agent would similarly increase their performance. In general, the addition of these tools makes it much more difficult to assess which of the three proposed novelties (test selection, extra tools, refinement loop) are eventually responsible for the superior performance. It seems that Tenet performs badly without all of them in place, but it seems non-obvious to me that e.g. a well configured OpenHands agent would not benefit from each of these independently. A more cleanly created ablation would make it easier to understand which components are crucial for future agent iterations.
 - The changes made to the default prompts of OpenHands, SWE-Agent and AutoCodeRover are not highlighted in the Appendix A, including for example the spurious addition to ignore any tests beyond the three provided tests for SWE-Agent. Please highlight all changes to the default prompt.

 I therefore think the paper is not ready for publication in the current state.

 Smaller issues:
 - The main evaluation should be done on at least on open-source LLM. This would allow comparing the results of Table 1 to the ablation results in the other Tables, in particular for related work. More crucially however, this helps both understand how well the findings generalize across different base models and makes the study reproducible (as closed-source models can be sunsetted).
 - The compared related works OpenHands and SWE-Agent appear to have lower performance. However, it should also be highlighted that these methods are fairly general. Concretely, they were designed for SWE-Bench, a benchmark of issue fixing to which Tenet can actually not be applied directly. Still, they obtain comparatively strong performance.
 - It seems weird that the variant without THM is provided the complete test suite (Line 318) and not three random test cases (as for the compared related works OpenHands and SWE-Agent). This should be made consistent.

 Nitpicks:
 - The overall Tenet seems very unagentic to me. I would expect Agents to somewhat decide on their own which actions to take when. The proposed method is more of a fixed pipeline that is potentially agentic in that some steps of the pipeline are executed by agents. This is overall not necessarily bad, similar work such as AgentLess [2] showed that completely unagentic pipelines can perform well on repository-level coding tasks as well. I would appreciate if this was clarified more in the paper.
- "vibe coding" (line 45) might be better referred to as "so-called 'vibe coding'"
- Line 348 uses citet instead of citep
- Please use section* for the reproduction statement (Line 489)
- Please provide a README describing how to set up, run and interpret results in the code in the supplementary submission
- The paper uses italics and boldface for entire sentences in several places in normal text. I personally think this looks unprofessional and hurts legibility, but opinions on this may diverge. IMO the better way to highlight core claims is to place them in paragraphs/titles.

[1] Mündler et al, SWT-Bench: Testing and Validating Real-World Bug-Fixes with Code Agents, NeurIPS 2024
[2] Xia et al, Agentless: Demystifying LLM-based Software Engineering Agents, FSE 2025

**Questions:**

Did you test if running the entire test suite led to timeouts (eg in ablation Table 3)? Are there timeouts for any agent actions? This is not outline in Section 4.

---

### Official Review · Reviewer_vYpc · 2025-10-29

**Soundness:** 2
**Presentation:** 2
**Contribution:** 2
**Rating:** 2
**Confidence:** 4

**Summary:**

This paper introduces TENET, an LLM agent designed for repository-level code generation that leverages Test-Driven Development (TDD). It argues that test cases can serve as crucial "executable specifications" to guide LLMs, overcoming the ambiguities of natural language instructions.

To solve the challenges of selecting useful tests, finding relevant code, and using test feedback, TENET features three key components:
1. A test harness mechanism to select a small but diverse suite of tests.
2. A tailored agent toolset for efficient code retrieval and interactive debugging.
3. A reflection-based refinement workflow to iteratively analyze test failures and refine the code.

Experiments demonstrate that TENET significantly outperforms existing agentic baselines on the REPOCOD and RepoEval benchmarks and concludes that a small, diverse set of test cases is often more effective than a large quantity for guiding code generation.

**Strengths:**

1. Novel and practically meaningful framing: The paper situates repository-level code generation within Test-Driven Development (TDD) and the emerging “vibe coding” paradigm, offering an executable-specification perspective that is both timely and impactful.

2. Substantial engineering contribution: The authors design a refined agent workflow and an augmented AST-based toolset, and provide thorough ablation studies. The scope and depth of implementation and evaluation reflect significant effort and rigor.

3. Improved effectiveness and efficiency: TENET achieves strong gains on benchmarks while reducing token consumption, demonstrating better agent efficiency without sacrificing accuracy.

**Weaknesses:**

1. Benchmark–scenario mismatch: The motivation—using TDD to address vibe coding—is compelling, but the chosen benchmarks (RepoEval, REPOCOD) diverge from realistic incremental development where repositories and unit tests are often incomplete and multi-file edits are required. These single-function, well-tested settings (RepoEval commonly used for fill-in-the-middle base-model evaluation) are simpler and less representative. Strong results on such benchmarks do not convincingly demonstrate real-world utility. The paper should evaluate on more practical tasks (e.g., SWE-bench) or provide IDE-based user studies to strengthen external validity.

2. Missing experimental details and fairness: The paper lacks key setup information (virtual environments, Docker build/run specifics for benchmarks, environment alignment across agents like OpenHands and SWE-Agent). It is unclear how TENET applies generated code without patches, whether all agents can view/execute unit tests equivalently, and what limits are set for RRW iterations. These omissions undermine reproducibility and the fairness of comparisons.

3. THM and RRW resemble prompt engineering: The proposed mechanisms can likely be emulated by existing agents via prompt or policy design (e.g., instructing test selection or repeated test-driven debugging). The paper should quantify how much of TENET’s gains persist when comparable prompts are added to OpenHands/SWE-Agent to validate that benefits stem from architectural novelty rather than prompt tuning.

4. Limited generality of the AST-based toolset: The tailored APIs are Python-centric and may not generalize to multi-language repositories, whereas terminal-based approaches exhibit broader language portability. The paper should discuss or test cross-language applicability.

5. Token cost analysis is incomplete: While TENET reduces input tokens, its output tokens are significantly higher than SWE-Agent, and output tokens typically cost more. This suggests TENET may not have a clear cost advantage relative to SWE-Agent; a more comprehensive cost accounting (wall-clock time, compute, total token spend) is needed.

**Questions:**

1. Could you report TENET’s performance on more realistic benchmarks such as SWE-bench Verified, and discuss whether the observed gains persist? Including these results would strengthen external validity.

2. How is the experimental environment set up (e.g., Docker/virtual environments, repository builds), and how is fairness ensured across agents (identical access to tests, time/compute budgets, tool availability)? Additionally, how do TENET’s modules interact during execution?

3. If THM and RRW requirements are added via prompts/policies to OpenHands or SWE-Agent (e.g., test selection and iterative test-driven debugging), do they yield similar improvements? How does their performance compare to TENET under matched conditions?

4. How does TENET perform on multilingual repository settings, such as SWE-bench Multilingual, and what adaptations (if any) are required for non-Python ecosystems?

---

### Official Review · Reviewer_ySXr · 2025-10-31

**Soundness:** 2
**Presentation:** 3
**Contribution:** 2
**Rating:** 4
**Confidence:** 4

**Summary:**

This paper introduces a code agent for function generation in repository-level scenarios under the test-driven development. It covers a test harness mechanism for selecting concise test suites to maximum the converage, a toolset the provide efficient retrieval, and a refinement workflow for iteratively reflecting errors during generation. This agent finally outperforms the sota methods and provides many findings behind the systematic experiments.

**Strengths:**

1. the workflow is easy to follow by framework figure.

2. this coverage-based test mechanism can enhance the quality of final delivered code.

3. results demonstrate the effectiveness of this agent, especially in REPOCOD dataset.

4. authors claimed that it is the first study of test-driven code generation in repo-level context.

**Weaknesses:**

1. It highly relys on the pre-defined test cases of each task.

2. Efficiency evaluation seems unfair. The API call limitation is set to 50 for SWE-Agent and Openhand, and 15 for TENET. It is better to analyze what the influence of number of API calls in baselines.

3. It is unclear that how does the agent perform on sparse or messy tests scenarios.

4. Sensitivity to the number of tests potentially shows that the agent is not robust to suite size in real-world.

**Questions:**

Please refer to the weaknesses section.

---

### Note · Authors · 2025-11-24

**Comment:**

We respectfully request to withdraw this submission. All co-authors have been informed and agreed with the withdrawal.

**Withdrawal Confirmation:**

I have read and agree with the venue's withdrawal policy on behalf of myself and my co-authors.